The Cryosphere Discuss., doi:10.5194/tc-2015-222, 2016 Manuscript under review for journal The Cryosphere Published: 15 January 2016 © Author(s) 2016. CC-BY 3.0 License.

This discussion paper is/has been under review for the journal The Cryosphere (TC). Please refer to the corresponding final paper in TC if available.

# Brief communication: On area- and slope-related thickness estimates and volume calculations for unmeasured glaciers

# W. Haeberli

Geography Department, University of Zurich, Winterthurerstrasse 190, 8057 Zurich, Switzerland

Received: 4 December 2015 - Accepted: 10 December 2015 - Published: 15 January 2016

Correspondence to: W. Haeberli (wilfried.haeberli@geo.uzh.ch)

Published by Copernicus Publications on behalf of the European Geosciences Union.

# Abstract

Area- and slope-related techniques have been used to estimate thicknesses and to calculate volumes of unmeasured glaciers on the basis of glacier outlines and corresponding glacier surface areas in glacier inventories. The present communication

- <sup>5</sup> critically reflects key aspects involved with the application of these approaches to field data. Area-related empirical statistics are known to only provide order-of-magnitude estimates if applied to individual glaciers or glacier ensembles spanning less than several orders of magnitude. Even at this scale, however, problems exist with respect to calibration/validation, error propagation, artefacts (immediate mass loss in case of coa-
- <sup>10</sup> lescing/disintegrating composite glaciers) and shortcomings (no detection of ice below sea level or below lake levels on land in view of glacier contributions to sea-level rise). 3-D-flux/stress/slope-related approaches and numerical models are better constrained by calibration/validation with field measurements. They help with overcoming the problems of 2-D-area-related statistics in that they allow for calculating detailed glacier bed
- topographies at all scales, from individual glaciers to global ensembles. Corresponding results are available today and can be further improved.

#### 1 Introduction

Area-related thickness estimates and corresponding volume calculations for unmeasured glaciers have been used for decades (UNESCO, 1970; Müller et al., 1977). As

a number of recent publications show (Andreassen et al., 2015; Martín-Español et al., 2015; Zekollari and Huybrechts, 2015), they are still frequently applied today. Such procedures, however, involve a number of shortcomings and the input data are limited to 2-D (planar) information, while alternatives using 3-D-topography (elevation, slope) have long been available (Haeberli and Hoelzle, 1995), more recently made striking
 progress in their application to DEM information as combined with glacier inventories (e.g., Huss and Hock, 2015) and provide more promising approaches. In particular,

the possibility to model detailed glacier-bed topographies reaches far beyond coarse and highly uncertain values of "average thicknesses" or "total volumes" estimated by area-related approaches. Furthermore, average basal shear stresses used with, or calculated from, flux/elevation/slope-related approaches provide a robust and transparent possibility to test the plausibility of calculated values. The following briefly outlines key

aspects involved concerning scatter and volume-area self-relation in statistical regression, area definition, error propagation, calibration/validation and limitation to 2-D information and average thicknesses. Model inter-comparison is recommended and the full use of available 3-D information is advocated.

#### Glacier areas, thicknesses and volumes 2 10

Glacier volumes V (unit:  $m^3$ ) are calculated from information about glacier thickness obtained from numerical models or determined in the field using drillings or geophysical soundings at points or profiles and inter-/extrapolated, averaged or integrated over measured glacier areas A (unit:  $m^2$ ). Corresponding technical procedures are described in recent studies by Andreassen et al. (2015) or Martín-Español et al. (2015). 15 A modern database of glacier thicknesses and areas is available from the World Glacier Monitoring Service (WGMS; Gärtner-Roer et al., 2014). Glacier volumes determined using field measurements contain the defined glacier areas from which they have been calculated:  $V = A \cdot h$ , where h = mean glacier thickness (unit: m) over the defined area (cf. the definition in Cogley et al., 2011).

#### Area-related approaches 3

20

25

The use of 2-D- (planar) area-related statistics for estimating thicknesses and volumes of unmeasured glaciers is still common. In particular, the direct statistical correlation between glacier volumes and glacier areas (often called "volume-area scaling") has become popular and is quite commonly considered to be the most simple and