# Peer review of "Brief communication: On area- and slope-related thickness estimates and volume calculations for unmeasured glaciers"

_The Cryosphere, 2015_

## Referee Comment (RC1) · Anonymous Referee #1 · 22 Feb 2016

The manuscript by Haeberli is a contribution to the ongoing discussion on methods of ice thickness estimation of glaciers without thickness measurements. Two camps are involved in this discussion: advocates for estimating ice volume estimates based on scaling with area, and advocates for using more complex procedures usually involving more geometric (and sometimes, climatic) information for obtaining spatially distributed ice thickness estimates.

Both sides of the debate have valuable arguments: there is little doubt that in principle, methods relying on a more detailed geometric representation of glaciers (i.e., slope, flux/stress-related parameters) are able to produce more accurate estimates of ice thickness. There is also little doubt that for large ensembles of glaciers, scaling

approaches lead to similarly reliable estimates of total ice volume. But unfortunately, both sides of the debate tend to overstretch their arguments: In the case of scaling-based estimates, it is the claim that the exponents of the scaling laws are firmly based in physics (see, e.g., the "width closure" in Bahr et al., 1997, which is purely empirical). In the case of slope/stress/...-related estimates, it is the claimed magnitude of improvements: I.e., the claimed ability to calculate "detailed glacier bed topographies at all scales"(Haeberli, in the manuscript on hand). It's a given that this is possible to calculate, but the uncertainties at small scales are necessarily immense.

Given this background, and as detailed below, I am not convinced that the manuscript represents substantial progress beyond the current scientific understanding. I do not see any arguments that have not been extensively discussed in the literature, and I also do not see a new assessment of the claimed vast superiority of the flux/stress/slope-related approaches at small scales (again, I am convinced they do provide better estimates for individual glaciers – but I am not convinced that they are able to robustly detect features in the bedrock that are an order of magnitude or more smaller than the glacier scale). I would also like to point out the IACS working group on ice thickness estimation, which is in the process of providing an assessment of all kinds of methods. Instead of repeating arguments and claims, it would perhaps be better to wait for the outcome of that project to renew the debate.

Detailed remarks:

- P2L8-10 (also further below): These shortcomings are rather related to the problematic application of VA-scaling to glaciers in a strong disequilibrium, leading to problems similar to the necessity to determine an "apparent mass balance" (Farinotti et al., 2009) for flux-based estimates.

- P2L12-13: This claim is not well justified. As stated above, I am convinced that flux/stress/slope-related approaches are able to outperform VA scaling, but not at all spatial scales. More importantly, as they rely on the same field measurements (of ice
thickness), but have more degrees of freedom, I would argue their calibration is in fact less well constrained.

- P3L1 (and similarly throughout the paper): "the possibility to model detailed glacier-bed topographies" does not imply that these topographies are very accurate. See, e.g., Fig. 8 in Linsbauer et al. (2012): I estimate that the typical error of their method is of the order of 50 % of the measured ice thickness (similar, e.g., in Farinotti et al., 2009, or Frey et al., 2014, their Fig. 8).

- P3L4-5: "provide a robust and transparent possibility to test the plausibility of calculated values" – yes, plausible – but not necessarily correct (my comment above).

- Section 2: it should at least be mentioned here that the uncertainty entering the volume measurements through inter- and extrapolation may be substantial as well.

- Section 3: The paper referenced in the beginning of this section (Bahr et al., 2015) gives a much more detailed account of all the different shortcomings of problematic applications of VA scaling than the section itself. This is understandable, but questions the necessity for this manuscript.

- P5L4-18: The more or less identical argument can be made for flux/stress/slope-related approaches: while all validations show that ice thickness estimates at small scales (i.e., within one glacier) have errors of about 50 % of the ice thickness, this limitation is often misunderstood or even ignored: in the literature, these approaches are frequently used to determine small scale bedrock features (such as overdeepenings, etc.).

- P5L19-24: It is not the correlation between glacier volumes and area that adds new information, but the ice volume measurements that determine the scaling parameters.

- P6L10-12: "Related predictive equations [...] essentially calculate glacier area from itself" – I disagree. It is obvious that there is information on ice thickness in ice area. Just as it is obvious that the correlation between volume and area must be greater than

between thickness and area.

- Sections 3.3. and 3.4: These shortcomings have almost identical effects no matter what method for ice thickness estimation you use – they are shortcomings of the field data, not of the methods. Regarding the overdeepenings: see above.

- P9L10: "quite detailed and realistic glacier-bed topographies": yes, they do look nice – but the details should mostly disappear within the error bars; see the validation figures in the references.

- P9L20-23: This is exactly the point! And it does not surprise me: All the stress/slope/flux related methods need to apply some kind of spatial filtering of the surface topography (and if it is only to overcome the problem of inverted slopes); the detailed shape of the bedrock will depend strongly on this filtering (and on errors in the DEM, etc.). The author is (justly!) criticizing claims (or at least, applications) that invoke the impression that VA scaling works for individual glaciers. But then, through-out the manuscript, he himself is invoking the impression that stress/slope/flux related methods are able to determine small-scale bedrock features - which the papers he references, and in his own text here, show to be just as problematic.

References:

Bahr, D. B., Meier, M. F., & Peckham, S. D. (1997). The physical basis of glacier volume?area scaling. Journal of Geophysical Research: Solid Earth, 102(B9), 20355-20362.

Farinotti, D., Huss, M., Bauder, A., Funk, M., & Truffer, M. (2009). A method to estimate the ice volume and ice-thickness distribution of alpine glaciers. Journal of Glaciology, 55(191), 422-430.

Linsbauer, A., Paul, F., & Haeberli, W. (2012). Modeling glacier thickness distribution and bed topography over entire mountain ranges with GlabTop: Application of a fast and robust approach. Journal of Geophysical Research: Earth Surface, 117(F3).

---

## Referee Comment (RC2) · Anonymous Referee #2 · 2 Mar 2016

This is a welcomed opinion paper by W. Haeberli, who contributes to an ongoing, relevant debate on whether volume-area regression is a robust method to estimate global glacier volume changes and ultimately to estimate the contribution from glaciers to sea-level rise. The viewpoints of the author are well-known to followers of The Cryosphere (cf. the interactive discussion of Frey et al.: The Cryosphere, 8, 2313-2333, 2014), but it is an advantage for all participants in the discussion and for the scientific community in general, to have the argumentation in an opinion paper, and a "Brief communication" in The Cryosphere is the best venue we have at the moment for expressing our scientific opinions in a peer-reviewed form.

While the paper is well-written, I find the emphases on area-related and slope-related

ice thickness estimates and volume calculations very unbalanced. Five pages have been devoted to area-related approaches, whereas only three less-structured pages focus on slope-related approaches. I recommend that the author tries to balance the discussion of the approaches more equally by using a similar sub-section structure for both sections.

The paper is best when the author presents strong arguments for the various short-comings of volume-area regression, whereas the argumentation is weaker, or at least somewhat ambiguous, in the appraisal of the flux/stress/slope-related approaches. Here, I am particular concerned about readers who want to familiarize themselves with the debate and the different approaches without reading 20+ other papers. The flux/stress/slope-related approaches should be appropriately introduced by briefly explaining what data is needed, how data is obtained, what the uncertainties are, differences between various flux/stress/slope-related approaches, and how these approaches estimate global glacier volume (Huss and Farinotti, 2012; Huss and Hock, 2015).

The uncertainties of the variables used in flux/stress/slope-related approaches should be addressed in a similar degree of detail as the variables for the area-related approaches. Check that there is complete consistency in the argumentation: If an argument is used to criticize shortcomings in area-related approaches, then make sure that the very same argument cannot be used to criticize shortcomings in flux/stress/slope-related approaches. Is slope "measured" or inter-/extrapolated on unmeasured glaciers? Do the uncertainties related to slope vary with glacier size and glacier morphology? How is the validation conducted on cirque and valley glaciers (should the maximum elevation be measured from the edge of the bergschrund or from the top of hanging ice on the headwall)? What is the robustness of flux and elevation range estimates of unmeasured small glaciers (<1 km2)?

DEMs are mentioned on page 8, line 25, but it is not explained, how they are combined with slope-related approaches. The shortcomings of DEMs must be presented in relation to unmeasured glaciers and the aim to estimate global glacier volume. These shortcomings include spatial resolution for small, remote glaciers (<1 km2), vertical resolution and inter-/extrapolations.

I also find some inconsistencies in other discussions of shortcomings. For example, problems with flat firn divides and debris-cover glaciers (page 6, line 20) should also be discussed for flux/stress/slope-related approaches. Also, it is sometimes unclear to the reader whether the discussion concerns the global population of glaciers or individual glaciers.

Anonymous Referee #1 has provided a thorough review and I support her/his reflections with regards to overstretching of arguments and the differences in performance of both approaches at different spatial scales. I recommend that the focus of this opinion paper is on thickness estimates and volume calculations for a global ensemble of glaciers (with the aim to estimate global glacier volume changes), and the arguments for using flux/stress/slope-approaches in this context need to be strengthened while the shortcomings of the two methods are discussed in a balanced way.

Some minor comments and suggestions:

Abstract:

- P2, L6-11: Avoid using unsupported statements in an Abstract. Instead, explain the aim and relevance of this opinion paper, the fundamentals of the two approaches, the shortcomings of both methods, and the recommendation of using flux/stress/slope-related approaches.

- P2, L12-14 (and throughout the paper): Be consistent in the naming of the approaches.

Introduction:

- My personal guess is that the apparent success of volume-area regression is partly due to the method's appealing simplicity. I think that this issue should be recognized in
the Introduction.

- P3, L3-5: This statement should be supported by references.

Area-related approaches:

- P4, L12-13 (here and elsewhere): The errors related to ground/airborne field measurements of ice thicknesses should be explained in section 2. The text becomes biased if they are only mentioned in relation to area-related approaches.

- P4, L17: Insert a reference to support the statement that "corresponding correlations are often weak for glaciers < 10 km2.

- P5, L3-18: What is the error of ice thickness (or volume) estimates based on area-related approaches for individual glaciers actually? What is the error of flux/stress/slope-related approaches? I am missing references to data here. It is not sufficient to write "... only at order-of-magnitude precision" without any references to actual error assessments.

- Sections 3.3 and 3.4 are on ice thickness data and should be moved to section 2 (or to a separate section on ice thickness data after section 2).

- P7, L25-26: Avoid unreferenced sentences such as this one, which undermines the argumentation. Why is it particularly important that "inter-/extrapolated thickness data [is] not "measured" but are products calculated using sometimes unknown or questionable assumptions"? Why is this argument not used with regards to slope determination or other variables? Is it really relevant to mention that inter-/extrapolated data is not measured? Which unknown or questionable assumptions are you referring to, and why is this statement not supported by references?

- P8, L1-3: This sentence is confusing. I am not sure what the author actually refers to here, and why he thinks that this is "quite serious". Why not discuss the robustness of various algorithms in interpolations along lateral glacier margins, if this is what is meant? It is unclear what "logistic reasons" refer to. Is it just with respect to ground-
based GPR measurements or does this postulate include airborne measurements as well? Why are there not any references, which support this argument?

- P8, L7: Section 3.5 should focus on estimation of global glacier volume, not the ice thickness estimate of individual glaciers. First the author makes it clear to readers that area-related approaches should not be used on individual glaciers or glacier ensembles spanning less than several orders of magnitude (P2, L7-8); and then the author "applies" area-related approaches on individual glaciers (Sections 3.4 and 3.5).

Slope-related approaches:

- See my earlier comments on structure, consistency in argumentation, lack of discussion about the determination of slope and elevation range, and errors in DEMs.

- P9, L7: The "various levels of complexity" need more attention. If models are very complex, they may become less "transparent" (P3, L4) for most potential users.

- P9, L15-17: Try to keep the focus in Section 4 on ". . . thickness estimates for all glaciers in the world . . ." and remove the "noisy" parts about overdeepenings and local ice thickness values.

- P9, L18: What is meant by "local"?

- P9, L20-25: It is difficult to evaluate the vague postulate that "the uncertainty of absolute thickness values seems to be lower than with area-related estimates but still remains high", when the author only presents uncertainties for the slope-related approaches "on average about +/-30 % for individual glaciers". Readers need to see comparative uncertainties in percentage for the area-related approaches.

- P9, L28-29: Discuss this "large number of assumptions about glacier mass balance and flow" in more detail.

- P10, L2: I guess that the average uncertainty is "+/- 5-10 %", not "+/- 5-10 m".

- P11, L6: "elevation is less sensitive to changes or misinterpretations of area". In order

to reach to this conclusion, there must be a discussion about how elevation is determined on unmeasured glaciers, what the uncertainties are, and the potential problems with hanging ice on headwalls, debris-covered/dead-ice proglacial areas, partly detached glacier termini, and interrupted glaciers.

- P11, L25-26: The information about the submission history seems unnecessary.

- Table and Figures: The captions are very long. I suggest that all captions are shortened.

- Figure 2: For consistency, change "Mean" to "Average" on the y-axis. Why is the y-axis intercept different from zero for curve B?

- Figure 2 caption: "Note the relation between average shear stress and elevation range and probably also with mass balance gradients". Are these relationships statistically significant? Move this sentence to the body text and show the statistics.

---

## Author Comment (AC1) · 3 Apr 2016

Area- and slope-related ice thickness estimates for unmeasured glaciers: plea and discussion

The following continues the discussion on area- and slope-related thickness estimates and volume calculations for unmeasured glaciers. It provides general comments on the submitted TCD Brief Communication and the feedback by the two anonymous referees. A complete version with a point-by point reply to the referee reports is provided in the attached PDF. The main goal is to make the various opinions and scientific arguments publicly accessible.

[Figure]

The intention of the submitted Brief Communication was to enable a publicly accessible discussion concerning widely used approaches for estimating ice thicknesses and calculating ice volumes on the basis of glacier inventory data. As such, this Brief Communication cannot be an extensive review but aims at being a plea for

- making a clear distinction between scaling theory and empirical regression/correlation statistics used for quantitative estimates;

- carefully considering the origin and nature of the field data (thicknesses, areas) used in such statistics;

- being careful about the use of area-area self regression/correlation in volume-area statistics for existing glaciers;

- comparing available approaches concerning scatter/uncertainties, error propagation, calibration/validation and applicability of products;

- using available 3D-input information to go beyond "average thicknesses" or "total volumes";

- recognizing the fascinating and important potential of such 3D input information for modeling detailed glacier-bed topographies in view of emerging research fields related to climate change impacts in cold regions, resulting environment/landscape dynamics, hazard conditions, land-use options and corresponding adaptation strategies, etc.

The two anonymous referees are thanked for their substantial feedback and contribution to the discussion. The constructive suggestions of Referee #2 show in what direction a detailed and systematic assessment with a more extended format than that of a short communication could go. The submitted Brief Communication can remain in its original form as a discussion or opinion paper (perhaps even a "wake-up call"). It had been elaborated over years in close communication with a number of colleagues. Together with the feedback from the two referees and the reply to these feedbacks it hopefully encourages further critical reflection. This is in support of the timely IACS

initiative mentioned by Referee #1. The fact that TCD offers a forum for such publicly accessible contributions to ongoing discussions is highly appreciated.

Concerning two aspects the view expressed in the Brief Communication clearly differs from the opinions and recommendations of the two referees: (1) the way scientific arguments have been dealt with in the literature (Referee #1) and (2) the safety and importance of numerically modeled glacier bed topographies (both referees).

(1) Without exact references, it is difficult to understand how Referee #1 cannot " … see any arguments that have not been extensively discussed in the literature …". The primary reason for publishing the Brief Communication was the fact that the essential aspects (bullet points) mentioned above still need to be adequately reflected in the scientific literature. The Brief Communication for the first time deals with these technical aspects as related to basic principles of both approaches in direct comparison. It may be seen as a complement to the quantitative inter-comparison and analysis of calculated results provided by Frey et al. (2014).

(2) The scientific arguments used by Referee #1 concerning numerically modeled glacier bed topographies (especially overdeepenings) must be more carefully considered. The robustness of spatial patterns in calculated bed topographies such as sequences of overdeepenings and riegels is not a direct function of the uncertainty related to ice thickness estimates. Spatial patterns and topologies of calculated glacier beds much more directly depend on surface slope than absolute ice thickness values calculated from highly uncertain assumptions about mass fluxes. The scientific evidence from dense radio-echo sounding networks (Gabbi et al., 2012) and exposed glacier beds (Haeberli and Linsbauer, 2013) is beyond doubt: even with strong mismatches of average ice thicknesses or total glacier volumes, spatial patterns of calculated bed topographies remain robust. Successful prediction is possible – within limits, of course – of sites with potential lake formation in overdeepenings, which become exposed by glacier retreat. In fact, spatial patterns of glacier bed topographies are safer and the corresponding knowledge more advanced (cf. Haeberli et al., 2016a, b; Linsbauer et

al., 2016) than assumed in the report of Referee #1. This also means that such bed topographies represent far more than "noise" in comparison with total global glacier volumes as suggested by Referee #2. The importance of such local to regional and even global applications of glacier inventory data and DEMs should be emphasized even more strongly.

References

Frey, H., Machguth, H., Huss, M., Huggel, C., Bajracharya, S., Bolch, T., Kulkarni, A., Linsbauer, A., Salzmann, N., and Stoffel, M.: Estimating the volume of glaciers in the Himalaya- Karakoram region using different methods, The Cryosphere, 8, 2313–2333, doi:10.5194/tc- 8-2313-2014, 2014. Gabbi, J., Farinotti, D., Bauder, A. and Maurer, H.: Ice volume distribution and implications on runoff projections in a glacierized catchment, Hydrology and Earth System Sciences 16, 4543–4556, doi:10.5194/hess-16-4543-2012, 2012. Haeberli, W. and Linsbauer, A.: Brief communication "Global glacier volumes and sea level – small but systematic effects of ice below the surface of the ocean and of new local lakes on land", The Cryosphere, 7, 817–821, doi:10.5194/tc-7-817-2013, 2013. Haeberli, W., Linsbauer, A., Cochachin, A., Salazar, C. and Fischer, U.H.: On the morphological characteristics of overdeepenings in high-mountain glacier beds, Earth Surface Processes and Landforms (accepted for publication), 2016b. Haeberli, W., Schaub, Y. and Huggel, C.: Increasing risks related to landslides from degrading permafrost into new lakes in de-glaciating mountain ranges, Geomorphology, doi:10.1016/j.geomorph.2016.02.009. 2016a. Linsbauer, A., Frey, H., Haeberli, W., Machguth, H., Azam, M. F., and Allen, S.: Modelling glacier-bed overdeepenings and possible future lakes for the glaciers in the Himalaya– Karakoram region, Annnals of Glaciology, 57, 119–130, doi:10.3189/2016AoG71A627, 2016.

Please also note the supplement to this comment:
http://www.the-cryosphere-discuss.net/tc-2015-222/tc-2015-222-AC1-supplement.pdf

[Figure]

**Supplement:**

**Area- and slope-related ice thickness estimates for unmeasured glaciers: plea and discussion**

**Reply to referee comments by Wilfried Haeberli**

The following continues the discussion on area- and slope-related thickness estimates and volume calculations for unmeasured glaciers. It starts with general comments on the submitted TCD Brief Communication and the feedback by the two anonymous referees before providing a point-by point reply to the referee reports and a final remark. The main goal is to make the various opinions and scientific arguments publicly accessible.

**General comments**

The intention of the submitted Brief Communication was to enable a publicly accessible and citable discussion concerning widely used approaches for estimating ice thicknesses and calculating ice volumes on the basis of glacier inventory data. As such, this Brief Communication cannot be an extensive and carefully equilibrated review but aims at being a *plea for*

- making a clear distinction between scaling theory and empirical regression/correlation statistics used for quantitative estimates;
- carefully considering the origin and nature of the field data (thicknesses, areas) used in such statistics;
- being careful about the use of area-area self regression/correlation in volumearea statistics for existing glaciers;
- comparing available approaches concerning scatter/uncertainties, error propagation, calibration/validation and applicability of products;
- using available 3D-input information to go beyond "average thicknesses" or "total volumes";
- recognizing the fascinating and important potential of such 3D input information for modeling detailed glacier-bed topographies in view of emerging research fields related to climate change impacts in cold regions, resulting environment/landscape dynamics, hazard conditions, land-use options and corresponding adaptation strategies, etc.

The two anonymous referees are thanked for their substantial feedback and contribution to the discussion. The constructive suggestions of Referee #2 show in what direction a detailed and systematic assessment with a more extended format than that of a short communication could go. The submitted Brief Communication can remain in its original form as a *discussion or opinion paper (perhaps even a "wake-up call")*. It had been elaborated over years in close communication with a number of colleagues. Together with the feedback from the two referees and the reply to these feedbacks below it hopefully encourages further critical reflection. This is in support of the timely IACS initiative mentioned by Referee #1. The fact that TCD offers a

forum for such publicly accessible contributions to ongoing discussions is highly appreciated.

Concerning two aspects the view expressed in the Brief Communication clearly differs from the opinions and recommendations of the two referees: (1) the way scientific arguments have been dealt with in the literature (Referee #1) and (2) the safety and importance of numerically modeled glacier bed topographies (both referees). Comments on these two aspects are first given below before providing a point-by point reply to the referee reports.

(1) Without exact references, it is difficult to understand how Referee #1 cannot "... see any arguments that have not been extensively discussed in the literature ...". The primary reason for publishing the Brief Communication was the fact that the essential aspects (bullet points) mentioned above still need to be adequately reflected in the scientific literature. The Brief Communication for the first time deals with these technical aspects as related to basic principles of both approaches in direct comparison. It may be seen as a complement to the quantitative inter-comparison and analysis of calculated results by Frey et al. (2014).

(2) The scientific arguments used by Referee #1 concerning numerically modeled glacier bed topographies (especially overdeepenings) must be more carefully considered. The robustness of spatial patterns in calculated bed topographies such as sequences of overdeepenings and riegels is not a direct function of the uncertainty related to ice thickness estimates. Spatial patterns and topologies of calculated glacier beds much more directly depend on surface slope than absolute ice thickness values calculated from highly uncertain assumptions about mass fluxes. The scientific evidence from dense radio-echo sounding networks (Gabbi et al., 2012) and exposed glacier beds (Haeberli and Linsbauer, 2013) is beyond doubt: even with strong mismatches of average ice thicknesses or total glacier volumes, spatial patterns of calculated bed topographies remain robust. Successful prediction is possible - within limits, of course - of sites with potential lake formation in overdeepenings, which become exposed by glacier retreat. In fact, spatial patterns of glacier bed topographies are safer and the corresponding knowledge more advanced (cf. Haeberli et al. 2016a, b) than assumed in the report of Referee #1. This also means that such bed topographies represent far more than "noise" in comparison with total global glacier volumes as suggested by Referee #2. The importance of such local to regional and even global applications of glacier inventory data and DEMs should be emphasized even more strongly.

**Point-by-point reply to the referee reports**

*Note: Reply comments below are in italics. Repetitions are unavoidable but mark essential points.*

Anonymous Referee #1

Received and published: 22 February 2016

The manuscript by Haeberli is a contribution to the ongoing discussion on methods of ice thickness estimation of glaciers without thickness measurements. Two camps are involved in this discussion: advocates for estimating ice volume estimates based on scaling with area, and advocates for using more complex procedures usually involving more geometric (and sometimes, climatic) information for obtaining spatially distributed ice thickness estimates.

Both sides of the debate have valuable arguments: there is little doubt that in principle, methods relying on a more detailed geometric representation of glaciers (i.e., slope, flux/stress-related parameters) are able to produce more accurate estimates of ice thickness.

**Agreed. This is one of the main messages of the text. Moreover, such methods also provide richer and more interesting information concerning various important applications of results.**

There is also little doubt that for large ensembles of glaciers, scaling approaches lead to similarly reliable estimates of total ice volume.

This remains to be discussed on the basis of careful reflection and inter-comparison of approaches/results as encouraged in the text. The Brief Communication primarily scrutinizes the underlying concept of area-related thickness estimates and volume calculations, the accuracy of resulting ice volume estimates being a secondary aspect.

But unfortunately, both sides of the debate tend to overstretch their arguments: In the case of scaling-based estimates, it is the claim that the exponents of the scaling laws are firmly based in physics (see, e.g., the "width closure" in Bahr et al., 1997, which is purely empirical).

Agreed. A clear distinction must be made between theoretical concepts and empirical techniques. Quantitative estimates of average glacier thicknesses or total glacier volumes are not undertaken with any (scaling) theory but use simple, fully empirical techniques of statistical regression/correlation. These empirical statistical techniques have been used long before they were called "scaling" and are independent of any theory. The origin and nature of the field data used in such statistics must be carefully examined. Volumes of existing glaciers are not independent of the corresponding glacier areas as assumed in volume-area scaling theory but calculated from them (there is no method to directly "measure" glacier volumes in the field, independently of quantitative thickness and area information). Straightforward application of volume-area scaling theory to field data on glaciers therefore leads to a relation between glacier areas as used for volume calculation and themselves. This is immediately evident from the unrealistic degrees of correlation (for instance,  $R^2 = 0.99$  in Bahr et al., 1997) as explained in the text of the Brief Communication (page 5, line 29; see also the explanations in the caption to Figure 1).

In the case of slope/stress/. . .-related estimates, it is the claimed magnitude of improvements: I.e., the claimed ability to calculate "detailed glacier bed topographies at all scales" (Haeberli, in the manuscript on hand). It's a given that this is possible to calculate, but the uncertainties at small scales are necessarily immense.

Referee #1 later also doubts the possibility "... to robustly detect features in the bedrock that are an order of magnitude or more smaller than the glacier scale" and thereby especially emphasizes the example of bed overdeepenings. This must be more carefully considered. First of all, the involved uncertainties are not "immense" but can be quantified as described in the text (page 9, lines 18 – page 10, line 3). Even more importantly, the calculations can be directly compared to detailed bed topographies documented by rapidly increasing numbers of radio-echo soundings and to bed-topographies exposed by retreating glaciers. True, the estimated depth values still remain highly uncertain as discussed in the text. The topology, however,

given by the spatial patterns of ice thicknesses and related bed surfaces is much more robust (cf. Gabbi et al., 2012; see especially their figures 10 and 11). This is due to the fact that such bed topologies primarily relate to surface topography and surface slope as defined by DEMs (Haeberli et al., 2016b). Probably the simplest and most striking example is the fact that bed overdeepenings only occur where glacier surface slopes are < 5 to  $10^{\circ}$  (cf. Frey et al., 2010). Even though the morphometry of the corresponding overdeepenings cannot be exactly calculated, their location and approximate size was successfully predicted in various cases of new lakes recently forming in the Swiss Alps (Gauli, Rhone, Trift, Kühboden, etc.; cf. Haeberli and Linsbauer, 2013). The modeled overdeepenings of a potential future lake at Glaciar Artesonraju in the Cordillera Blanca or of the enlarging Imja Tsho in the Mount Everest region is in close agreement with results from radio-echo soundings (Linsbauer et al., 2016, Somos-Valenzuela et al., 2015; and unpublished data). Cases like Trift, Kühboden, Artesonraju or Imja Tsho relate to changing hazard conditions in cold mountains, especially concerning impact waves from large rock falls originating in steep icy slopes surrounding new lakes due to ice de-buttressing and permafrost degradation (Haeberli et al., 2016a). Flux/stress/slope-related approaches in such cases already today provide an important knowledge basis for long-term planning. Their scientific and practical potential should be critically evaluated but not be underestimated or even generally disqualified.

Given this background, and as detailed below, I am not convinced that the manuscript represents substantial progress beyond the current scientific understanding. I do not see any arguments that have not been extensively discussed in the literature,

This statement without references is difficult to understand. As mentioned in the text, some aspects have indeed been described in the literature before. The problem of composite glaciers in area-related statistics is an example. Other aspects have not been treated adequately (for instance, self-correlation and corresponding artefacts in volume-area statistics) or have not even been mentioned at all (for instance, error propagation in area-related thickness estimates and volume calculations). The important option to make use of quantitative information from glacier fore-fields becoming increasingly exposed as a consequence of glacier retreat are hardly recognized so far and a consideration of basal shear stresses for plausibility checks cannot be found in the literature. Most importantly, however, the present contribution is the first attempt at systematically and critically reflecting the advantages, limitations and perspectives for the future of both, area- and flux/stress/slope-related approaches. This is complementary to the quantitative inter-comparison of calculated results provided by Frey et al. (2013, 2014) for the Himalaya-Karakoram region.

... and I also do not see a new assessment of the claimed vast superiority of the flux/stress/slope- related approaches at small scales (again, I am convinced they do provide better estimates for individual glaciers – but I am not convinced that they are able to robustly detect features in the bedrock that are an order of magnitude or more smaller than the glacier scale).

This was discussed above. "Total volumes" are not the only important or interesting product of ice thickness estimates for unmeasured glaciers. High-resolution bed topographies as bases for studies about possible future landscape dynamics in cold regions related to climate change impacts, adaptation strategies, options for future land-use, changing hazard conditions or melt-water paths to the ocean, for instance, are important and useful products in their own right. As key elements in emerging research fields they are of a scientific and practical significance, which reaches far beyond anything area-related estimates of total glacier volumes could possibly provide.

I would also like to point out the IACS working group on ice thickness estimation, which is in the process of providing an assessment of all kinds of methods. Instead of repeating arguments and claims, it would perhaps be better to wait for the outcome of that project to renew the debate.

The IACS initiative is most welcome, thanks for the information. It illustrates the importance of, and need for, critically reflecting and assessing the available approaches and their potential products and perspectives. The submitted text together with the feedbacks from the two referees and the present reply may be seen as an external input to this timely initiative.

Detailed remarks:

- P2L8-10 (also further below): These shortcomings are rather related to the problematic application of VA-scaling to glaciers in a strong disequilibrium, leading to problems similar to the necessity to determine an "apparent mass balance" (Farinotti et al., 2009) for flux-based estimates.

*True*. *This important aspect is mentioned for flux/stress/slope-related approaches on page 10, lines 6-7, but indeed also affects area-related approaches.*

- P2L12-13: This claim is not well justified. As stated above, I am convinced that flux/stress/slope-related approaches are able to outperform VA scaling, but not at all spatial scales.

The question about the added value produced by area-related approaches remains open. The problem of composite glaciers – representing most if not all of the larger glaciers on Earth – seems to be a severe limit concerning the reliability of arearelated estimates even at global scales. In their final conclusions, for instance, Bahr et al. (2015) recommend: " ... Avoid applying volume-area scaling to glacier complexes. Do not apply volume-area scaling to individual parts or branches of a glacier." To what sort of glaciers should the approach then be applied? And even more generally: for what reason would the limitation to 2D information as input constitute an advantage over the use of the available 3D information?

More importantly, as they rely on the same field measurements (of ice thickness), but have more degrees of freedom, I would argue their calibration is in fact less well constrained.

Yes, this is true for flux-driven slope-related approaches (Farinotti et al., 2009, Huss and Farinotti, 2012, Huss and Hock, 2015), which parameterize all components of the surface mass balance and glacier flow. As these parameterizations involve many degrees of freedom and large uncertainties, such approaches must be heavily tuned and calibrated using local ice thickness information from radio-echo soundings, etc. The much simpler and straightforward stress-driven approaches (Linsbauer et al., 2009, 2012, 2016) use an empirical relation between average shear stress and elevation range for each glacier. They need no tuning and can be directly compared to local ice thickness information from radio-echo soundings, etc. Area-related approaches provide "average thicknesses", which cannot be directly compared to local point or profile information about ice thickness from field observations.

- P3L1 (and similarly throughout the paper): "the possibility to model detailed glacier- bed topographies" does not imply that these topographies are very accurate. See, e.g., Fig. 8 in Linsbauer et al. (2012): I estimate that the

typical error of their method is of the order of 50 % of the measured ice thickness (similar, e.g., in Farinotti et al., 2009, or Frey et al., 2014, their Fig. 8).

**Agreed. This is discussed on page 9, lines 22-26, which also provides the references to Linsbauer et al. (2012), Farinotti et al. (2009) and Frey et al. (2014).**

- P3L4-5: "provide a robust and transparent possibility to test the plausibility of calculated values" – yes, plausible – but not necessarily correct (my comment above).

Agreed. This should be self-explanatory with the use of the term "plausibility".

- Section 2: it should at least be mentioned here that the uncertainty entering the volume measurements through inter- and extrapolation may be substantial as well.

Volumes of existing glaciers cannot be "measured" but are always calculated from quantitative thickness and area information. Uncertainty aspects are indeed important and are treated on page 9, line 18 to page 10, line 3.

- Section 3: The paper referenced in the beginning of this section (Bahr et al., 2015) gives a much more detailed account of all the different shortcomings of problematic applications of VA scaling than the section itself. This is understandable, but questions the necessity for this manuscript.

Bahr et al. (2015) as cited in the text indeed provide a thorough discussion of volumearea scaling theory and – in the abstract of their review paper – specifically mention problems with numerous applications in the scientific literature "... that are in conflict with the theory ...". Especially interesting is their recommendation to apply volume-area scaling only to an individual glacier if the result is treated as an order of magnitude estimate, and to avoid applying volume-area scaling to glacier complexes and to individual parts or branches of a glacier. In addition to such important restrictions and recommendations from the side of scaling theory, emphasis is here given on complementary technical-practical aspects related to empirical area-related regression/correlation. Such technical-practical aspects concern the origin and nature of used field data, the resulting statistical self-correlation with its artefacts, the problem of error propagation with area-related thickness estimates and volume calculations, the difficulty of calibrating "average thicknesses" or "total volumes" with point or profile observations of ice thickness and the limitations related to the use of the resulting "average thicknesses" and "total volumes", especially concerning sea-level studies.

- P5L4-18: The more or less identical argument can be made for flux/stress/slope- related approaches: while all validations show that ice thickness estimates at small scales (i.e., within one glacier) have errors of about 50 % of the ice thickness, this limitation is often misunderstood or even ignored: in the literature, these approaches are frequently used to determine small scale bedrock features (such as overdeepenings, etc.).

Extreme deviations of slope-related model calculations from local ice thickness information can even be higher than 50%. Average deviations for large glacier ensembles are closer to 30% as explained on page 9, lines 20-25. The robustness of calculated bed features such as overdeepenings, however, does not depend on these thickness uncertainties alone but must be compared with the robustness of spatial patterns and bed topologies as explained above. Even with a systematic ice depth deviation of 30 to 50% the spatial pattern of a modeled bed surface can be robust and realistic. This can be tested by model inter-comparison or using field measurements. Gabbi et al. (2012; se especially their figures 10 and 11), for instance, document a large volume difference of 36% between numerically modeled ice thicknesses and results from a dense radio-echo sounding network in a catchment of the Swiss Alps. At the same time, their study clearly confirms the robustness of the spatial patterns in the modeled bed topographies.

- P5L19-24: It is not the correlation between glacier volumes and area that adds new information, but the ice volume measurements that determine the scaling parameters.

Again, glacier volumes cannot be "measured" but are calculated from thickness and area information. The mentioned scaling parameters therefore concern a relation between glacier volumes and the glacier areas from which these glacier volumes had been calculated. Does this make sense? Correspondingly and as explained on page 6, lines 4-9, the extreme  $R^2$ -values (>0.9) of the resulting statistical correlations are artefacts from statistical self-correlation, which provides a false impression about the origin and quality of the used field data. This should honestly be made clear.

- P6L10-12: "Related predictive equations [. . .] essentially calculate glacier area from itself" – I disagree. It is obvious that there is information on ice thickness in ice area. Just as it is obvious that the correlation between volume and area must be greater than between thickness and area.

The question is much simpler: area is on both sides of the corresponding predictive equation 1 on page 6. In the same way,  $R^2$ -values of volume-area relations are extremely high simply because area is used in both variables and, hence, compared to itself (page 6, lines 4-9).

- Sections 3.3. and 3.4: These shortcomings have almost identical effects no matter what method for ice thickness estimation you use – they are shortcomings of the field data, not of the methods.

The error propagation mentioned here relates to the fact that calculations of glacier volumes from area-related statistics combine areas and their uncertainties with average thicknesses estimated from the same areas and their uncertainties. In area-related volume calculations, therefore, the area-related uncertainty enters twice. Flux/stress/slope-related approaches decouple thickness estimates from area and, hence, avoid this error propagation effect. Section 3.3 makes this clear.

Slope-related calculations of ice thicknesses and bed topographies can be directly and quantitatively compared with results from field measurements of ice thickness at points or profiles or with bed topographies in glacier fore-fields exposed by glacier retreat. As explained in section 3.4, this is not the case for average thicknesses or total volumes estimated or calculated using area-related statistics.

Sections 3.3 and 3.4 point to a basic difference between area- and slope-related approaches concerning error propagation and calibration/validation. The case of glacier fore-fields becoming exposed through glacier retreat illustrates the corresponding perspectives. Figure 2 in the text is an example for the still hardly recognized fact that glacier fore-fields becoming exposed through ongoing glacier retreat represent a rich and important source of quantitative information on glacier geometries. This information can be used for further development and improvement of slope-related approaches for estimating thicknesses and glacier bed topographies of unmeasured glaciers. Glacier fore-fields becoming exposed do not document "average glacier thicknesses" or "total volumes" of glaciers as estimated by arearelated statistics but reveal detailed glacier bed topographies which can be directly compared to glacier-bed topographies modeled using flux/stress/slope-related approaches. Such comparison documents that realistic modeling of glacier overdeepenings and potential lake formation is possible already now (cf. Haeberli and Linsbauer, 2013) and can further be improved.

Regarding the overdeepenings: see above.

The question of overdeepenings has been dealt with above: it is true that further improvements are necessary and possible. The robustness of spatial ice thickness patterns must thereby be considered rather than the uncertainty of individual ice depth values alone. This robustness primarily depends on surface slope as given by DEMs and can be tested by model inter-comparison, field observations and comparison with exposed glacier fore-fields. Experience from such exercises (Gabbi et al., 2012, Haeberli and Linsbauer, 2013) and comparison with morphological indicators (Frey et al. 2010) documents the success of corresponding model calculations.

- P9L10: "quite detailed and realistic glacier-bed topographies": yes, they do look nice – but the details should mostly disappear within the error bars; see the validation figures in the references.

As repeatedly explained above, the spatial patterns of modeled ice thicknesses and bed surfaces are quite robust despite the uncertainty in absolute ice thickness values: a calculated bed topography can systematically be at too low or too high elevation but nevertheless remain robust and realistic in its spatial pattern and topology.

- P9L20-23: This is exactly the point! And it does not surprise me: All the stress/slope/flux related methods need to apply some kind of spatial filtering of the surface topography (and if it is only to overcome the problem of inverted slopes); the detailed shape of the bedrock will depend strongly on this filtering (and on errors in the DEM, etc.). The author is (justly!) criticizing claims (or at least, applications) that invoke the impression that VA scaling works for individual glaciers. But then, through- out the manuscript, he himself is invoking the impression that stress/slope/flux related methods are able to determine small-scale bedrock features - which the papers he references, and in his own text here, show to be just as problematic.

Sure, applied filtering techniques must be carefully considered and clearly defined as done, for instance, by Linsbauer et al., 2012). Further improvements and reduction of uncertainties are needed and possible, especially in combination with observations in nature. The calculation of detailed bed topographies, however, is not only robust and successful (within limits, of course) already today at local to global scales but offers perspectives for future developments and important environment- and policy-related applications, which reach far beyond estimates of "average thicknesses" or "total volumes" from area-related statistics. The IACS working group will hopefully recognize the significance of such emerging fields of research and application in glacier science.

**References:**

Bahr, D. B., Meier, M. F., & Peckham, S. D. (1997). The physical basis of glacier volume?area scaling. Journal of Geophysical Research: Solid Earth, 102(B9), 20355- 20362.

Farinotti, D., Huss, M., Bauder, A., Funk, M., & Truffer, M. (2009). A method to estimate the ice volume and icethickness distribution of alpine glaciers. Journal of Glaciology, 55(191), 422-430. Linsbauer, A., Paul, F., & Haeberli, W. (2012). Modeling glacier thickness distribution and bed topography over entire mountain ranges with GlabTop: Application of a fast and robust approach. Journal of Geophysical Research: Earth Surface, 117(F3).

**Anonymous Referee #2**

Received and published: 2 March 2016

This is a welcomed opinion paper by W. Haeberli, who contributes to an ongoing, relevant debate on whether volume-area regression is a robust method to estimate global glacier volume changes and ultimately to estimate the contribution from glaciers to sea-level rise. The viewpoints of the author are well-known to followers of The Cryosphere (cf. the interactive discussion of Frey et al.: The Cryosphere, 8, 2313-2333, 2014), but it is an advantage for all participants in the discussion and for the scientific community in general, to have the argumentation in an opinion paper, and a "Brief communication" in The Cryosphere is the best venue we have at the moment for expressing our scientific opinions in a peer-reviewed form.

**Thanks. This exactly corresponds to the wish expressed by a number of colleagues (including experts in glacier inventory work) to make an open discussion publicly accessible.**

While the paper is well-written, I find the emphases on area-related and slope-related ice thickness estimates and volume calculations very unbalanced. Five pages have been devoted to area-related approaches, whereas only three less-structured pages focus on slope-related approaches. I recommend that the author tries to balance the discussion of the approaches more equally by using a similar sub-section structure for both sections.

**This is a useful suggestion in view of a more extended, systematic assessment.**

The paper is best when the author presents strong arguments for the various shortcomings of volume-area regression, whereas the argumentation is weaker, or at least somewhat ambiguous, in the appraisal of the flux/stress/slope-related approaches. Here, I am particular concerned about readers who want to familiarize themselves with the debate and the different approaches without reading 20+ other papers. The flux/stress/slope-related approaches should be appropriately introduced by briefly explaining what data is needed, how data is obtained, what the uncertainties are, differences between various flux/stress/slope-related approaches, and how these approaches estimate global glacier volume (Huss and Farinotti, 2012; Huss and Hock, 2015).

Agreed. This will be essential in a more extended and systematic assessment but is beyond the scope of a Brief Communication. The following short summary of flux/stress/slope-related approaches is taken from the supplementary material to Haeberli et al. (2016b):

Three primary approaches presently exist:

- (1) Stress-driven models (Linsbauer et al. 2009, 2012; Paul and Linsbauer 2012; Frey et al., 2014) assume a constant shear stress for each individual glacier as defined by an empirical relation between average basal shear stress and glacier elevation range governing mass turn-over (Haeberli and Hoelzle, 1995). This approach is simple and empirical but has the advantage of only requiring easily available input information and of being transparent, robust and fast in its application.
- (2) Flux-driven models (Farinotti et al., 2009, Huss and Farinotti, 2012) parameterize a complex set of processes to simulate surface mass flux and

corresponding flow of ice; basal shear stresses are not assumed a priori but produced by such models and can vary within individual glaciers. This elegant approach involves a comprehensive process understanding. Many of the involved processes can, however, not precisely be parameterized. Heavy tuning is therefore necessary, which reduces the pursued complexity and makes the approach quite empirical again.

(3) Combined flux/stress-driven models (Clarke et al., 2013) combine the approaches (1) and (2). They optimize the use of all available information (mass flux, shear stress) but leave reduced possibilities for independent comparison with the approaches (1) and (2).

The uncertainties of the variables used in flux/stress/slope-related approaches should be addressed in a similar degree of detail as the variables for the area-related approaches. Check that there is complete consistency in the argumentation: If an argument is used to criticize shortcomings in area-related approaches, then make sure that the very same argument cannot be used to criticize shortcomings in flux/stress/slope-related approaches.

Again agreed. This will be essential in a more extended and systematic assessment. An example is the problem of ill-defined areas (see page 6, lines 19-20), which affects area- and flux/stress/slope-related approaches in a contrasting way: uncertainties in area definition affect ice thickness over the entire glacier surface in area-related approaches but only in the marginal area of uncertainty in flux/stress/slope-related calculations. Another effect is due to mistakes in geo-referencing and corresponding misplacement of glacier outlines, which strongly affect flux/stress/slope-related but not area-related calculations. This second effect will likely reduce with future improvements in the quality of geo-referencing and in the resolution/reliability of DEMs.

Is slope "measured" or inter-/extrapolated on unmeasured glaciers? Do the uncertainties related to slope vary with glacier size and glacier morphology? How is the validation conducted on cirque and valley glaciers (should the maximum elevation be measured from the edge of the bergschrund or from the top of hanging ice on the headwall)? What is the robustness of flux and elevation range estimates of unmeasured small glaciers (<1 km2)?

**Yes, these are indeed all important points to be dealt with in a more extended and systematic assessment.**

DEMs are mentioned on page 8, line 25, but it is not explained, how they are combined with slope-related approaches. The shortcomings of DEMs must be presented in relation to unmeasured glaciers and the aim to estimate global glacier volume. These shortcomings include spatial resolution for small, remote glaciers (<1 km2), vertical resolution and inter-/extrapolations.

Right. These are all important technical aspects. Detailed comments can be found in the corresponding papers (for instance, Huss and Farinotti, 2012; Frey et al., 2014, Huss and Hock, 2015; Linsbauer et al., 2016). Regional aspects can thereby be the primary goal rather than global glacier volumes.

I also find some inconsistencies in other discussions of shortcomings. For example, problems with flat firn divides and debris-cover glaciers (page 6, line 20) should also be discussed for flux/stress/slope-related approaches.

Problems related to flat firn divides and debris-covered glacier tongues for flux/stress/slope-related approaches are mentioned on page 10, lines 6-7 and can be more fully discussed in a systematic review.

Also, it is sometimes unclear to the reader whether the discussion concerns the global population of glaciers or individual glaciers.

As explained in the text, the use of area-related statistics should be limited to large glacier ensembles spanning several orders of magnitude. This means that they should – if at all – primarily be used at global scale or in regions such as the Karakoram, which include very large glaciers. Flux/stress/slope-related approaches are interesting at local, regional and global scales concerning small as well as large glaciers and samples of glaciers.

Anonymous Referee #1 has provided a thorough review and I support her/his reflections with regards to overstretching of arguments and the differences in performance of both approaches at different spatial scales. I recommend that the focus of this opinion paper is on thickness estimates and volume calculations for a global ensemble of glaciers (with the aim to estimate global glacier volume changes), and the arguments for using flux/stress/slope-approaches in this context need to be strengthened while the shortcomings of the two methods are discussed in a balanced way.

A balanced discussion including all scientific arguments can be attempted in a more extended systematic review paper. One important emphasis in the Brief Communication is on the fact that flux/stress/slope-related approaches provide important results at various scales and enabling applications beyond "total global glacier volumes".

Some minor comments and suggestions:

Abstract:

- P2, L6-11: Avoid using unsupported statements in an Abstract. Instead, explain the aim and relevance of this opinion paper, the fundamentals of the two approaches, the shortcomings of both methods, and the

Which statements are "unsupported"? Of course, the text could be written in a different style. This, however, should rather be the task of an extended, more systematic and correspondingly more extended review paper.

- P2, L12-14 (and throughout the paper): Be consistent in the naming of the approaches.

Agreed. The somewhat lengthy but exact term "flux/stress/slope-related approach" is used throughout this reply text.

Introduction:

- My personal guess is that the apparent success of volume-area regression is partly due to the method's appealing simplicity. I think that this issue should be recognized in the Introduction.

Exactly: The extreme simplicity of empirical volume-area regression is one of the keys to understand the amazing popularity of this approach. In many cases the so-called volume-area scaling indeed seems to have been applied because "it is simple and everybody does it". Another key to understand the popularity of this strongly limited, empirical approach is the transformation – decades ago already – of weak thickness-area relations into seemingly good-looking volume-area self-regression/correlation with corresponding scatter plots and predictive equations. This step made the factor "ice thickness" to disappear and the awareness concerning the origin and nature of field data to weaken (cf. Frey et al., 2013 and the corresponding interactive reviews/discussions in TCD). A third key is the straightforward connection of this empirical self-regression/correlation with the volume-area scaling theory as based on continuum mechanics (Bahr et al. 1997). The

artefacts ( $R^2$ -values > 0.9) produced by such statistics together with the use of the theory-related term "volume-area scaling" (instead of area-related regression/correlation) may have created the impression that a reliable, precise, sophisticated and generally accepted technique was used. In reality, many applications are technically problematic (self-correlation, error propagation, results of limited use) and even in conflict with scaling theory (Bahr et al., 2015).

- P3, L3-5: This statement should be supported by references.

This aspect is discussed on page 10, lines 8-27 where a number of references are given.

Area-related approaches:

- P4, L12-13 (here and elsewhere): The errors related to ground/airborne field measurements of ice thicknesses should be explained in section 2. The text becomes biased if they are only mentioned in relation to area-related approaches.

True. Possible errors in field measurements also affect flux/stress/slope-related approaches and can be discussed in detail by a more extended and systematic review. Emphasis here, however, is on slope effects as an important source of scatter in straightforward area-related statistics. Inclusion of slope effects is possible in multiple regression/correlation (Grinsted, 2013) and indeed seems to provide more plausible results (see page 10, lines 25-28). With such inclusion of slope in arearelated statistics, however, the question arises why the applied slope information is not directly used to calculate area-independent flux/stress/slope-related ice thicknesses (this would help avoiding the problems of statistical self-correlation and error propagation).

- P4, L17: Insert a reference to support the statement that "corresponding correlations are often weak for glaciers < 10 km2.

Bahr et al. (2015) and Andreassen et al. (2015) as cited in the text (page 6, lines 6-9) are good sources. See especially the squared correlation coefficients provided there.

- P5, L3-18: What is the error of ice thickness (or volume) estimates based on area-related approaches for individual glaciers actually? What is the error of flux/stress/slope-related approaches? I am missing references to data here. It is not sufficient to write ". . . only at order-of-magnitude precision" without any references to actual error assessments.

The uncertainty range of average ice thicknesses is discussed on page 4, lines 9-20 for values estimated by area-related regression and on page 9, lines 20-25 for flux/stress/slope-related ice thickness calculations.

- Sections 3.3 and 3.4 are on ice thickness data and should be moved to section 2 (or to a separate section on ice thickness data after section 2).

This could be a possibility. Emphasis is here on the use of ice thickness data from field measurements in area-related statistics. The use of ice thickness data from field measurements in flux/stress/slope-related approaches is discussed on page 9, lines 18-25.

- P7, L25-26: Avoid unreferenced sentences such as this one, which undermines the argumentation. Why is it particularly important that "inter-/extrapolated thickness data [is] not "measured" but are products calculated using sometimes unknown or questionable assumptions"? Why is this argument not used with regards to slope determination or other variables? Is it really relevant to mention that inter-/extrapolated data is not measured? Which unknown or questionable assumptions are you referring to, and why is this statement not supported by references?

This should be common sense and may need no reference. Average thicknesses used in area-related statistics have been produced by inter-/extrapolation of point or profile information over entire glacier areas. This is sometimes done "by hand" or with "expert knowledge", sometimes using a constant basal shear stress approximation (!) or by applying mathematical software (cf. Gärtner-Roer et al., 2014 and the corresponding ice thickness database of WGMS). Glacier-bed topographies obtained from flux/stress/slope-related calculations can be directly compared to point or profile information and even with quantitative information from exposed glacier beds. Examples of such procedures have been given by, for instance, Farinotti et al. (2009), Linsbauer et al. (2012), Haeberli and Linsbauer (2013) or Frey et al. (2014).

- P8, L1-3: This sentence is confusing. I am not sure what the author actually refers to here, and why he thinks that this is "quite serious". Why not discuss the robustness of various algorithms in interpolations along lateral glacier margins, if this is what is meant? It is unclear what "logistic reasons" refer to. Is it just with respect to ground-based GPR measurements or does this postulate include airborne measurements as well? Why are there not any references, which support this argument?

The robustness of various inter-/extrapolation techniques as mentioned above should indeed be carefully discussed (cf. Gärtner-Roer et al., 2014; or the inter-comparison of bed-overdeepenings calculated with various DEMs and interpolation schemes provided in figure 2 of Haeberli et al., 2016b). One basic problem relates to the fact that information from drillings and geophysical soundings tends to be much richer and better for flat, crevasse-free glacier sections with compressing flow, relatively thick ice and often over-deepened bed topography. This is not only due to limits of accessibility concerning measurements at the glacier surface but also to scattering effects in airborne radio-echo soundings caused by the presence of crevasses. The difficulty of inter-/extrapolating good information from flat/thick glacier parts to less well-documented steeper/thinner and often heavily crevassed glacier sections is likely to be responsible for a considerable part of the uncertainty concerning all thickness estimates for entire glaciers.

- P8, L7: Section 3.5 should focus on estimation of global glacier volume, not the ice thickness estimate of individual glaciers. First the author makes it clear to readers that area-related approaches should not be used on individual glaciers or glacier ensembles spanning less than several orders of magnitude (P2, L7-8); and then the author "applies" area-related approaches on individual glaciers (Sections 3.4 and 3.5).

This may be a misunderstanding. Area-related estimates of average glacier thicknesses and total glacier volumes for large glacier ensembles at regional to global scale use the outlines and corresponding areas of individual glaciers as contained in glacier inventories. Total glacier volumes are then calculated by summing up all average glacier thicknesses for all size classes using the regression equations from corresponding statistics based on field data. The main point is that such average ice thicknesses concerning size classes should not be applied to individual glaciers or only as an order-of-magnitude estimate. The use of information about individual glaciers, however, is unavoidable. Slope-related approaches:

- See my earlier comments on structure, consistency in argumentation, lack of discussion about the determination of slope and elevation range, and errors in DEMs.

**Thanks again. These useful comments have been treated above and should be carefully considered in a more extended and systematic review paper.**

- P9, L7: The "various levels of complexity" need more attention. If models are very complex, they may become less "transparent" (P3, L4) for most potential users.

True. Explanations on this point are given above in the reply about existing approaches. The trade-off concerning complexity and transparency is indeed an important point which should be carefully considered. Full flux-driven numerical models are complex and suitable for sensitivity studies concerning effects of the various parameterizations involved. Transparency, however, is much higher with simple-empirical stress-driven models. Such simple-transparent approaches contain a limited amount of process understanding but can have considerable advantages in practical applications.

- P9, L15-17: Try to keep the focus in Section 4 on "... thickness estimates for all glaciers in the world . . ." and remove the "noisy" parts about overdeepenings and local ice thickness values.

Thanks for this suggestion but emphasis should remain on the importance of goals beyond estimating total global glacier volumes, which can be reached by flux/stress/slope-related numerical models.

- P9, L18: What is meant by "local"?

**"Local" (point or profile information about ice depth or bed surface) is here understood as contrast to "average" (mean ice thickness over the entire glacier).**

- P9, L20-25: It is difficult to evaluate the vague postulate that "the uncertainty of absolute thickness values seems to be lower than with area-related estimates but still remains high", when the author only presents uncertainties for the slope-related approaches "on average about +/-30 % for individual glaciers". Readers need to see comparative uncertainties in percentage for the area-related approaches.

Exact comparison with point or profile data from field measurements is only possible for flux/stress/slope-related approaches. Comparison with area-related statistics can be made on the basis of scatter plots such as Figure 1 and 2 as explained in the text or as plausibility considerations with respect to values of basal shear stresses (page 10, lines 8-27). As explained in the text the uncertainty ranges for both approaches concerning ice thicknesses only differ moderately. Much more important are systematic deviations (for instance, related to neglected slope effects or to the problem with composite glaciers) and the type of information (average thickness versus bed topographies).

- P9, L28-29: Discuss this "large number of assumptions about glacier mass balance and flow" in more detail.

*This is done above in the reply concerning existing flux/stress/slope-related approaches.*

- P10, L2: I guess that the average uncertainty is "+/- 5-10 %", not "+/- 5-10 m".

- P11, L6: "elevation is less sensitive to changes or misinterpretations of area". In order to reach to this conclusion, there must be a discussion about how elevation is deter- mined on unmeasured glaciers, what the uncertainties are, and the potential problems with hanging ice on headwalls, debris-covered/dead-ice proglacial areas, partly de-tached glacier termini, and interrupted glaciers.

Yes, perfectly true and important. These aspects need systematic investigation. Some glaciers in the Himalaya-Karakoram region (Frey et al., 2014) with elevation ranges of 4 km include extremely steep icy walls with thin ice or hanging glaciers. Decisions on including or excluding such glacier parts in inventory work can affect thicknesses estimated from both approaches. Applying an upper limit of average basal shear stress as in GlabTop (Frey et al., 2014; Linsbauer et al., 2016) reduces the problem for glaciers with extreme elevation ranges but not for small glaciers in stress-driven flux/stress/slope-related approaches. The problem affects all glacier sizes in area-related approaches.

- P11, L25-26: The information about the submission history seems unnecessary.

To provide information about the submission history was a question of honesty in connection with the resubmission to TCD. A detailed explanation had been added for the TC editors. The Journal of Glaciology was informed about the resubmission and corresponding explanation.

- Table and Figures: The captions are very long. I suggest that all captions are shortened.

Thanks for the suggestion, which concerns a question of style. The intention here was to help with careful reflection of the graphs, which contain fundamentally important information.

- Figure 2: For consistency, change "Mean" to "Average" on the y-axis. Why is the y-axis intercept different from zero for curve B?

Thanks also for this suggestion, which is fully appropriate. The curve fitting for curve *B* could be improved in order to have a true zero intercept.

- Figure 2 caption: "Note the relation between average shear stress and elevation range and probably also with mass balance gradients". Are these relationships statistically significant? Move this sentence to the body text and show the statistics.

 $R^2$ -values vary between 0.28 for the Bernese Alps and 0.68 for South Grison Alps and Glarus Alps. Such values are comparable with the  $R^2$ -values for thickness-area regression related to similar glacier size categories.

**An outlook**

It will be interesting to see the future development in the field of quantitative thickness estimates for unmeasured glaciers.

Application of area-related statistics to estimate average thicknesses and to calculate total volumes of glaciers enabled first assessments of global glacier volumes but also produced an increasing number of highly variable regression or scaling parameters for smaller and smaller glacier ensembles spanning lower and lower orders of magnitudes. Extrapolation of this trend eventually leads to scaling parameters for individual glaciers – a product of questionable value and in conflict with the aims and principles of volume-area scaling theory.

The use of digital terrain information with flux/stress/slope-related numerical models for calculating distributed ice thicknesses and detailed glacier-bed topographies also enabled assessments of global glacier volumes but in addition opened new fields of glacier science. Further improvements are necessary and possible. Highly relevant applications related to research on climate change impacts and to corresponding policy-relevant environmental considerations are possible.

**References concerning the reply text**

- Clarke, G. K. C., Anslow, F. S., Jarosch, A. H., Radić, V., Menounos, B., Bolch, T., and Berthier, E.: Ice volume and subglacial topography for western Canadian glaciers from mass balance fields, thinning rates, and a bed stress model, J. Climate, 26, 4282–4303, doi:10.1175/JCLI-D-12-00513.1, 2013.
- Farinotti, D., Huss, M., Bauder, A., and Funk, M.: An estimate of the glacier ice volume in the Swiss Alps, Global Planet. Change, 68, 225–231, doi:10.1016/j.gloplacha.2009.05.004, 2009.
- Frey, H., Haeberli, W., Linsbauer, A., Huggel, C. and Paul, F.: A multi level strategy for anticipating future glacier lake formation and associated hazard potentials, Natural Hazards and Earth System Science 10, 339-352, 2010.
- Frey, H., Machguth, H., Huss, M., Huggel, C., Bajracharya, S., Bolch, T., Kulkarni, A., Linsbauer, A., Salzmann, N. and Stoffel, M.: Ice volume estimates for the Himalaya–Karakoram region: evaluating different methods, The Cryosphere Discuss., 7, 4813–4854, doi:10.5194/tcd-7-4813-2013, 2013.
- Frey, H., Machguth, H., Huss, M., Huggel, C., Bajracharya, S., Bolch, T., Kulkarni, A., Lins- bauer, A., Salzmann, N., and Stoffel, M.: Estimating the volume of glaciers in the Himalaya- Karakoram region using different methods, The Cryosphere, 8, 2313–2333, doi:10.5194/tc- 8-2313-2014, 2014.
- Gabbi, J., Farinotti, D., Bauder, A. and Maurer, H.: Ice volume distribution and implications on runoff projections in a glacierized catchment, Hydrology and Earth System Sciences 16, 4543–4556, doi:10.5194/hess-16-4543-2012, 2012.
- Grinsted, A.: An estimate of global glacier volume, The Cryosphere, 7, 141–151, doi:10.5194/tc-7-141-2013, 2013.

Haeberli, W. and Hoelzle, M.: Application of inventory data for estimating

characteristics of and regional climate-change effects on mountain glaciers: a pilot study with the European Alps, Ann. Glaciol., 21, 206–212, 1995.

- Haeberli, W. and Linsbauer, A.: Brief communication "Global glacier volumes and sea level – small but systematic effects of ice below the surface of the ocean and of new local lakes on land", The Cryosphere, 7, 817–821, doi:10.5194/tc-7-817-2013, 2013.
- Haeberli, W., Schaub, Y., Huggel, C.: Increasing risks related to landslides from degrading permafrost into new lakes in de-glaciating mountain ranges, Geomorphology, doi:10.1016/j.geomorph.2016.02.009. 2016a.
- Haeberli, W., Linsbauer, A., Cochachin, A., Salazar, C. and Fischer, U.H.: On the morphological characteristics of overdeepenings in high-mountain glacier beds, Earth Surface Processes and Landforms (accepted for publication), 2016b.
- Huss, M. and Farinotti, D.: Distributed ice thickness and volume of all glaciers around the globe, J. Geophys. Res., 117, F04010, doi:10.1029/2012JF002523, 2012.
- Huss, M. and Hock, R.: A new model for global glacier change and sea-level rise, Front. Earth Sci., 3, 32 pp., doi:10.3389/feart.2015.00054, 2015.
- Linsbauer, A., Paul, F., and Haeberli, W.: Modeling glacier thickness distribution and bed topog- raphy over entire mountain ranges with GlabTop: application of a fast and robust approach, J. Geophys. Res., 117, F03007, doi:10.1029/2011JF002313, 2012.
- Linsbauer, A., Frey, H., Haeberli, W., Machguth, H., Azam, M. F., and Allen, S.: Modelling glacier-bed overdeepenings and possible future lakes for the glaciers in the Himalaya– Karakoram region, Ann. Glaciol., 57, 119–130, doi:10.3189/2016AoG71A627, 2016.
- Linsbauer, A., Paul, F., Hoelzle, M., Frey, H., and Haeberli, W.: The Swiss Alps without glaciers – a GIS-based modelling approach for reconstruction of glacier beds, in: Proceedings of Geomorphometry 2009, Zurich, Switzerland, 31 August–2 September, 243–247, 2009.
- Paul F, Linsbauer A. 2012. Modeling of glacier bed topography from glacier outlines, central branch lines, and a DEM. International Journal of Geographic Information Science 26, 1173–1190, doi:10.1080/13658816.2011.627859.
- Somos-Valenzuela, M.A., McKinney, D.C., Rounce, D.R. and Byers, A.C.: Changes in Imja Tsho in the Mount Everest region of Nepal, The Cryosphere, 8, 1661–1671, doi:10.5194/tc-8-1661-204, 2014